# Compact GC-QEPAS for On-Site Analysis of Chemical Threats

**DOI:** 10.3390/s23010270

**Published:** 2022-12-27

**Authors:** Nicola Liberatore, Roberto Viola, Sandro Mengali, Luca Masini, Federico Zardi, Ivan Elmi, Stefano Zampolli

**Affiliations:** 1Consorzio CREO, 67100 L’Aquila, Italy; 2Institute for Microelectronics and Microsystems, Italian National Research Council CNR-IMM, 40129 Bologna, Italy; 3Prometehus srl., Via P. Gobetti 101, 40129 Bologna, Italy

**Keywords:** gas analyzer, MEMS GC, QEPAS, forensic, safety and security

## Abstract

This paper reports on a compact, portable, and selective chemical sensor for hazardous vapors at trace levels, which is under development and validation within the EU project H2020 “RISEN”. Starting from the prototype developed for a previous EU project, here, we implemented an updated two-stage purge and trap vapor pre-concentration system, a more compact MEMS- based fast gas-chromatographic separation module (Compact-GC), a new miniaturized quartz-enhanced photoacoustic spectroscopy (QEPAS) detector, and a new compact laser source. The system provides two-dimensional selectivity combining GC retention time and QEPAS spectral information and was specifically designed to be rugged, portable, suitable for on-site analysis of a crime scene, with accurate response in few minutes and in the presence of strong chemical background. The main upgrades of the sensor components and functional modules will be presented in detail, and test results with VOCs, simulants of hazardous chemical agents, and drug precursors will be reported and discussed.

## 1. Introduction 

In the framework of the EU Horizon 2020 project: Real-time on-site forensic trace qualification (RISEN) [1], the development of a network of sensors for deployment at the crime scene is foreseen. Here, we present a compact gas sensor for real-time analysis of vapors while entering the scene to protect the health and safety of forensic personnel. Moreover, the sensor will be used for spot analysis of vapors released by forensic evidence to provide chemical information complementary to other contactless proximal sensors. Analysis of vapors, in particular, might reveal what solvents, acids, or precursors have been used in a synthesis, this, in turn, being crucial information for tracking the origin of the evidence. Our gas analyzer for RISEN is an upgrade of the first GC-QEPAS sensor prototype developed for ROCSAFE [2], which was dedicated to on-site analysis of chemical threats onboard a small Robotic Ground Vehicle (RGV), in the aftermath of industrial accidents or CBRN events. The sensing scheme in ROCSAFE, which proved to be effective, has been maintained, but most of the components and sub-modules of the sensor have been updated, mainly to improve compactness and portability, reduce power consumption, and extend the spectral range. Concerning the sensor concept and sensing scheme, we refer to our previous paper [3] for details. It is useful to remember that the sensor consists of three main functional blocks: a gas sampling module based on commercial sorbent tubes to capture and concentrate gas/vapors from the air [4]; a MEMS-based gas-chromatographic module to separate the components of the sampled air mix [5]; a photoacoustic detection module based on QEPAS [6] for identification of the sampled vapors [7,8,9,10]. These three modules are hyphenated to achieve the selectivity and sensitivity required for analysis of real samples on site. To the best of our knowledge, these characteristics make GC-QEPAs competitive with both GC-MS instruments, which are bulkier and more complex due to operation in vacuum, and with Ion Mobility Spectroscopy systems, which are more prone to false alarm and memory effects. The following sensor description only details the main hardware upgrades of the RISEN prototype. The new sensor was assembled and validated with mixes of solvents, simulants of hazardous chemical agents, and drug precursors. Results from laboratory tests will be presented in the last paragraph.

## 2. Sensor Description

The complete GC-QEPAS sensing chain with upgraded functional modules is schematically shown in Figure 1 and consists of:

—A sampling and concentration unit based on a commercial stainless-steel tube packed with two different meshes of graphitized carbon sorbent to capture substances of different volatility;

—A Compact-GC module, including a MEMS-based second-stage pre-concentrator and an all MEMS-based separation unit, including the fast separation column and the injector [11];

—A QEPAS detector using a new tunable external cavity quantum cascade laser (EC-QCL) IR source and a new miniaturized absorption detection module (ADM) for QEPAS analysis, containing the piezoelectric quartz tuning fork.

**Figure 1 sensors-23-00270-f001:**
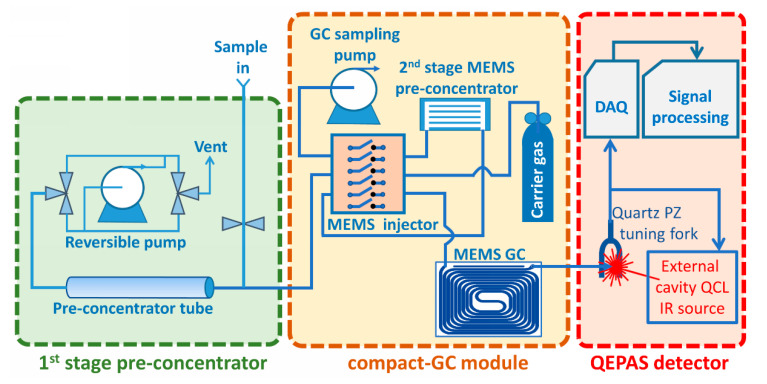
Sensing chain of the GC-QEPAS sensor upgraded for RISEN.

Figure 2 is a picture of the whole system, with the case lid removed and the three main modules in view. The naked sensor weighs 4 kg and 6.5 kg, including current metallic case. The power consumption is about 65 W in an idle state (ready to run) after a warmup phase, which lasts for about 5 min. The average power consumption during analysis is about 100 W with peaks of 170 W. It completes one measurement cycle (from sampling to response, including the time to complete the elution) in about 5 min, with a total energy need of approximately 8 Wh per cycle, and can be powered with a battery pack at 24 V. The main upgrades introduced in the RISEN GC-QEPAS sensor with respect to the ROCSAFE prototype [3] are detailed in the next paragraphs.

### 2.1. Sampling and Concentration Unit

It is based on a commercial stainless-steel sorbent tube from Markes, which was filled with two different meshes of graphitized carbon sorbent, namely Carbograph 5TD (Markes International Ltd., Llantrisant, UK) (specific surface area of 240 m^2^/g) and Carbograph 2TD (Markes International Ltd., Llantrisant, UK) (specific surface area of 10 m^2^/g) from Markes International Ltd., thus, allowing one to capture both volatile compounds (VOCs), such as acetone, alcohols, and solvents, as well as semi-volatile compounds (SVOCs), such as most persistent chemical agents and drug precursors. The main working parameters of the sampling and concentration unit are reported in Table 1. As shown in Figure 2, the sampling unit developed for RISEN is assembled in close connection with the compact-GC module inside the sensor case. This sampling and concentration unit allows one to sample the air at a flowrate of 0.85 L/min and to desorb the preconcentrated vapors into the second-stage MEMS-based pre-concentrator of the Compact-GC separation module. In the context of the experiments reported in this paper, the sampling and concentration unit was used for sampling from headspace of vials filled with liquids, or for the direct sampling of controlled air/vapor mixes prepared inside a 60-L glass box.

### 2.2. MEMS Compact-GC

Compact-GC is an innovative all-on-MEMS platform for the implementation of miniaturized or portable multi-purpose gas-chromatographic (GC) systems. The use of a directly interconnected MEMS technology for all three main GC components allows for a very efficient system integration and for a zero dead-volume and chemically inert analytical chain, which can be heated up to 200 °C to avoid sample condensation. In the compact GC fluidic circuit, a MEMS pre-concentrator is used as enriching sample loop and sample focuser, which is injected into the MEMS GC separation column by means of a MEMS injector based on an array of micromachined externally piloted membrane valves. The MEMS GC column consists of a stack between a 1 mm-thick Silicon wafer and a 0.5 mm-thick Pyrex wafer. The GC column channels are etched in the silicon wafer, which has platinum thin-film metallization (heater and temperature sensor) on the backside for temperature control. The Pyrex wafer has laser machined in- and outlet holes. The two wafers are anodically bonded and the overall size of the GC column is 25 × 25 × 1.5 mm^3^. The MEMS injector consists of a stack of three silicon wafers bonded using SU-8 photoresist as adhesive. One of the SU-8 layers is also used as membrane material for the five integrated externally piloted membrane valves. The use of MEMS technology results in very low consumption of power and consumables, with typical carrier gas consumptions, which is Nitrogen in our prototype, below 10 mL/min, which were carefully matched with the internal volume of the QEPAS detector cell. Further details about the compact-GC implementation can be found in [5]. Figure 3a shows a photograph of the compact-GC prototype used in this work, while Figure 3b shows a rendering of the fluidic core, high-lighting the three MEMS devices. The main working parameters of the MEMS GC module are reported in Table 2. The average power consumption of the whole module is 15 W with peaks of 60 W.

### 2.3. QEPAS Module

It utilizes a widely tunable external cavity quantum cascade laser IR source to interrogate vapors eluted from the MEMS GC module into a miniaturized ADM for QEPAS analysis, which includes a commercial quartz tuning fork (QTF) resonating at 32,768 Hz between two stainless-steel micro-resonator tubes (on-axis configuration). The main working parameters of the laser and of the QEPAS ADM are reported in Table 3.

#### 2.3.1. QEPAS QCL Source

A compact quantum cascade laser source from Block Engineering was implemented in the QEPAS analysis module. It is the OEM Mini-QCL-100 model shown in Figure 2, which is composed of a small laser optical head of less than 1 square inch, connected to its drive electronic boards by means of a short flat cable. This allows one to achieve a compact coupling with the optics and the miniaturized QEPAS cell. The assembly was integrated inside the sensor case, as shown in Figure 2. The laser can continuously scan the thermal IR spectrum in a wavelength range between 7.4 µm and 10.7 µm, to perform the spectroscopic analysis of each of the substances released sequentially by the Compact-GC module. In the context of the experiments reported in this paper, the laser was cycling scanned between 8 µm and 10 µm and taking ~2 s for each scan. The average power emitted in this spectral range is between 0.8 mW and 2 mW. Some modifications to the control software were implemented, in order to control the new QCL source properly. The laser is SOAP (Simple Object Access Protocol)-controlled, and a control program was developed to manage the laser in an automatic way.

#### 2.3.2. QEPAS Detector

A new miniaturized QEPAS analysis cell was developed and implemented. The cell is shown in Figure 4 close to a EUR 0.05 coin for size comparison. It is smaller than the one developed for ROCSAFE, allowing one to reduce the power and time needed for heating the cell to avoid recondensation of vapors of high boiling compounds inside. Moreover, the internal volume for analysis surrounding the quartz tuning fork was further reduced from 30 µL of the ROCSAFE QEPAS cell to 10 µL of actual RISEN cell. The reduced internal volume ensures a better match with the volumes of the chromatographic peaks (a few microliters) eluted by the Compact-GC, avoiding their broadening and remix, thus, retaining separation while they are flowing inside the QEPAS cell.

### 2.4. Electronics, Fluidics, Thermal Control, and Power Supply

Due to its high source impedance, the QTF is especially vulnerable to external noise, so the preamplifier stage was placed as close as possible to QTF and enclosed inside a shielded metal case. The main upgrade on the transimpedance amplifier (TIA) was the implementation of a composite amplifier scheme, as depicted in Figure 5, where a first FET input amplifier (AD8244 from Analog Devices Inc., Wilmington, MA, USA) is used as “buffer” for the second amplifier. When using a composite amplifier, it is important to pay attention to stability, meaning that the unity-gain crossover frequency of the operational amplifier must be less than the AD8244 bandwidth for this configuration to be unity-gain stable. Moreover, an alternative QEPAS amplifier topology based on voltage amplifier circuit using a differential amplifier [12] is under development.

Concerning the QCL control, a PCB was developed for generating the trigger pulses to drive the laser and a synchronous sine wave (with adjustable phase and amplitude) for the lock-in reference channel. A further PCB was developed for the acquisition of both the analog and digital signals coming from the laser. The digital signal allows one to monitor the trigger of the scan and the analog signal allows one to monitor the current wavenumber during the scan. Another upgrade of the RISEN prototype consists of the use of a different Single-Board Compute (SBC), model Lattepanda, which has smaller dimensions and lower power consumption (less than 5 W) than the Mini PC used in the ROCSAFE prototype. Another upgrade was a co-processor Arduino compatible (ATmega32u4), which was integrated into the SBC and used for controlling some features of the GC-QEPAS sensor, such as the Nitrogen input pressure, the internal temperature, and automatic control of the cooling fans.

Concerning the fluidics, the main upgrade was in pressure management, since the GC needs a first supply for the gas carrier of the GC column (Nitrogen at 0.5 bar) and a second supply for the actuation of the MEMS injector valves (Nitrogen at 2.0–2.5 bar). This was implemented using a small aluminum distribution block equipped with some miniature fittings (model M-5AU-4, from SMC), a pressure regulator (model AR10-M5H-1-A, from SMC), and an Absolute Integrated Pressure Sensor (model MPXH6400A, from NPX). The Pressure regulator was set to 0.4 Bar, allowing for a flux of approximately 5 sccm through the GC column. The pressure sensor was used to check the input pressure level. The analog output of theMPXH6400A was acquired from the microcontroller input inside the Lattepanda.

The new miniaturized QEPAS cell was heated using two mini ceramic rods (Ø 2 mm, 15 mm length, 30 W max), which can withstand temperatures up to 350 °C, as compared to the power film resistor used in the previous prototype, which was limited to 150 °C.

Concerning power management, in the preamplifier board and the other analog boards inside the sensor, no switching-mode power regulators were used, in order to avoid introducing their noise into the QEPAS system. To manage the voltages in the analog, QEPAS board linear regulators, such as ADP7142, are used, due to their high-power supply rejection, low noise, and excellent line and load transient response. The ADP7142 regulator output noise is 11 μV rms for the fixed options of 5 V or less.

## 3. Results and Discussion

In the framework of the RISEN Project, which is focused on the analysis of a crime scene, the GC-QEPAS sensor was tested in the lab with several volatile and semi-volatile compounds (VOCs and SVOCs), including simulants of toxic chemical agents and drug precursors, since the sensor will be dedicated to preserving the safety of operators entering the scene and to identify chemical traces. The main results of the experiments carried out are presented in the next paragraphs to demonstrate the capability of the GC-QEPAS sensor to retrieve the IR spectral fingerprints of different compounds in a mix, even if at trace level and in the presence of strong interferents. We developed an analysis software for our GC-QEPAS sensor to compare each acquired QEPAS spectrum with an internal spectral database by means of the Pearson correlation coefficient. Once the substance in the spectral database with the highest correlation is selected, a cross-check with the retention time of the GC separation is applied. Only if the spectral correlation exceeds 0.8 and the retention time is within the expected time window, a positive detection is reported. Using this algorithm, we obtained no false negatives and identified true positives. Since QEPAS spectra in the infrared spectral range are very similar to infrared absorption spectra, the standard reference databases from NIST and from PNNL were used successfully for identification. Therefore, the best results presented hereafter were obtained by using an experimental database of QEPAS spectra acquired previously with our GC-QEPAS sensor prototype.

### 3.1. Mix of Acetone and DPGME

We sampled air from the head space of a Becher filled with liquid acetone for 3 s, then we sampled for another 3 s from the headspace of another vessel filled with liquid DPGME (Di-propylene-glycol-methyl-ether), which is a simulant of the nerve agent Sarin. Looking at Figure 6a, two chromatographic peaks at ~45 s and ~110 s and the corresponding spectra of acetone (Figure 6b) and DPGME (Figure 6d) were clearly identified. Further, a third unexpected peak at ~80 s, whose spectrum corresponds to acetic acid (Figure 6c), was found. This could be explained by the partial conversion of acetone into acetic acid during first-stage pre-concentration, due to the desorption at high temperature (up to 300 °C), which takes place in the air, e.g., in the presence of oxygen and water vapor. In order to verify this assumption, we made two different experiments, sampling only vapors of acetone from headspace. In the first test, the desorption temperature of the sorbent tube was set to 280 °C, while in the second test, the desorption temperature was set to 180 °C, which is still suitable for desorption of such volatile compounds as acetone. The chromatograms obtained from the two experiments are reported in Figure 7. We obtained the same two peaks corresponding to acetone and acetic acid, but they exhibit comparable intensity only when desorbing at 280 °C (Figure 7a). On the contrary, they show very different intensities if desorbing at 180 °C, with the peak of acetone being almost doubled and a very small peak of acetic acid (Figure 7b). These results seem to confirm that the high temperature of desorption in air could promote partial conversion of acetone into acetic acid.

### 3.2. Mix of Gasoline and DMMP

In order to test the sensor performance with trace amounts of target compounds and in the presence of a strong background of interferents, we used a glass box of about 60 L internal volume. Inside the glass box, a hotplate is used to promote evaporation of small amounts of liquids injected through a septum by means of a microsyringe. To simulate a realistic and strong background of interferents, we introduced a few milliliters of liquid gasoline on a Petri dish inside the box and waited for saturation of the volume with vapors of gasoline. Then, we injected 1 microliter of liquid DMMP (dimethyl-methylphosphonate), which is another simulant of nerve agents, onto the hotplate for vaporization inside the box. If we ideally assume the total vaporization of the injected liquid, we can estimate an upper limit of the resulting concentration of DMMP of ~4 ppm. The air mix inside the box was analyzed after sampling with a flowrate of 850 mL/min for 60 s, resulting in a sample volume of 0.85 L. The result of the analysis is illustrated in Figure 8. The color map in Figure 8 is a 3D plot where the elution time is on the horizontal axis, the wavelength of acquired photoacoustic spectra is on the vertical axis, and the color bar is related to the intensity of the measured spectra. The color map shows how several components in the sample mix were resolved. In particular, the ‘Gasoline’ dashed curve encloses all components of gasoline vapors eluted between 50 s and 110 s, and the plot indicated by the red arrow shows the spectrum acquired at ca. 120 s corresponding to DMMP. From the signal-to-noise ratio (SNR) of the measured spectrum, a limit of detection (LoD) of 10 ppb was estimated for DMMP, which is almost four-times better than the result of 36 ppb obtained with the ROCSAFE prototype.

### 3.3. Drug Precursors: BMK

In the framework of the RISEN Project, the GC-QEPAS sensor will also be dedicated to the identification of chemical traces from evidence present on the scene. In particular, the sensing capability against drug precursors was tested for this purpose. We used the same procedure described in the previous paragraph for vaporizing 10 mL of the drug precursor BMK (benzyl-methyl-ketone), which is liquid at room temperature, inside the glass box of 60 L, and we sampled vapors of BMK for 60 s. The results are reported in Figure 9. Though we injected only BMK, both the color map (Figure 9a) and the plot of integral absorbance (Figure 9b) clearly show three different components. As for the tests with acetone, this could be explained by the partial decomposition of BMK, due to the high temperature of desorption in air of vapors captured in the first stage of pre-concentration. Looking at Figure 9a,b, a first peak was detected at ca. 80 s. It was identified as acetic acid by the correlation of the corresponding measured spectrum with the database (Figure 9c). A second peak was detected at ca.115 s, and it was identified as benzaldehyde by the correlation of the corresponding measured spectrum with the database (Figure 9d). Note that the elution time of benzaldehyde is close to the values observed in previous experiments for DPGME and DMMP, so the acquisition of the QEPAS spectrum was demonstrated to be essential for the identification. There was also a third peak detected at ca. 150 s, which was finally identified as BMK from the correlation of the corresponding measured spectrum with the database (Figure 9e).

### 3.4. Mix of Toxic Agent Simulants and Drug Precursors

A challenging mix of simulants of toxic agents plus one drug precursor was used to demonstrate the sensor identification capability against complex mixes. It was composed of DPGME, DMMP (simulants of nerve agent), methyl salicylate (simulant of blister agents), and safrole (precursor of amphetamine). All these compounds were sampled for a few seconds from headspace. The result of the analysis is reported in Figure 10, showing the color map (Figure 10a) and the plots of measured spectra with corresponding best fit from database allowing one to identify DPGME (Figure 10b), DMMP (Figure 10c), safrole (Figure 10d), and methyl salicylate (Figure 10e), respectively.

## 4. Conclusions and Outlook

A further development of the GC-QEPAS sensor, whose first prototype was developed for the Horizon 2020 EU Program within the project ROCSAFE, is ongoing in the project RISEN. Several upgrades, both on the GC module and the QEPAS module, have been implemented to approximately halve the size and weight of the sensor and to reduce power needs, in order both to improve man portability and promote utilization on robotic arms and onboard small UGVs (Unmanned Ground Vehicles) for on-site analysis. The main modification of the hardware was described, consisting of pushing towards the utilization of MEMS components, miniaturization of the QEPAS cell for analysis, compact design of the sub-assemblies, and increasing the working spectral range. Results from lab testing of the new prototype with some compounds of interest in the analysis of a crime scene were reported. In particular, fast analysis, with less than 5 min from sampling to response, and identification capability at ppb level of drugs’ precursors and simulants of toxic chemicals in the CBRNe context were demonstrated, even in the presence of interferents at much higher vapor concentrations. Further investigation for validation of the sensor performance by means of field trials is foreseen within RISEN in the next period.

## Figures and Tables

**Figure 2 sensors-23-00270-f002:**
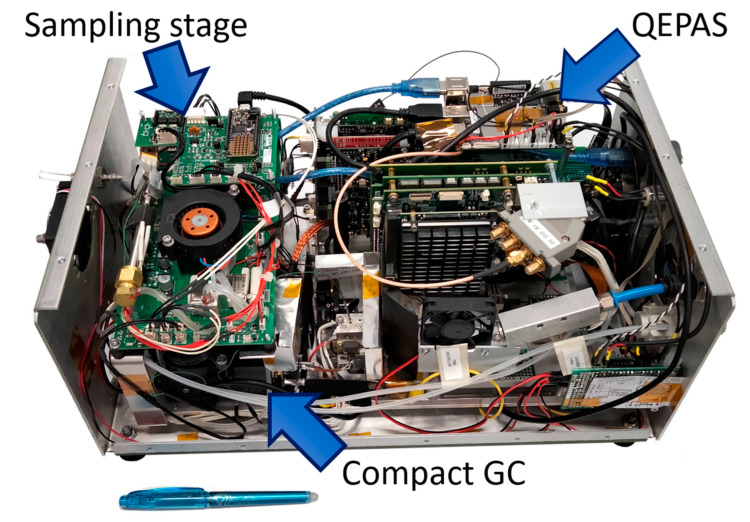
GC-QEPAS sensor assembled for RISEN.

**Figure 3 sensors-23-00270-f003:**
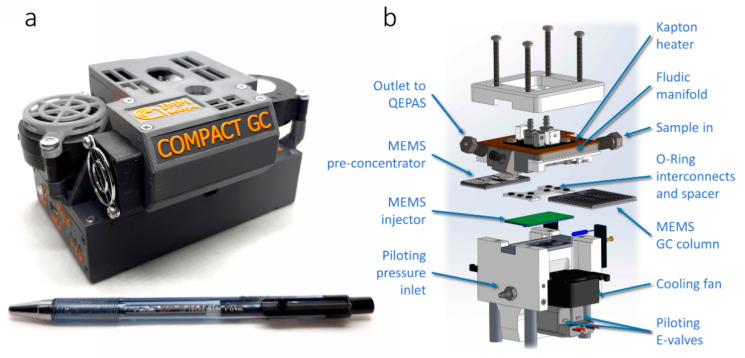
Compact-GC prototype (**a**) and rendering of core components (**b**).

**Figure 4 sensors-23-00270-f004:**
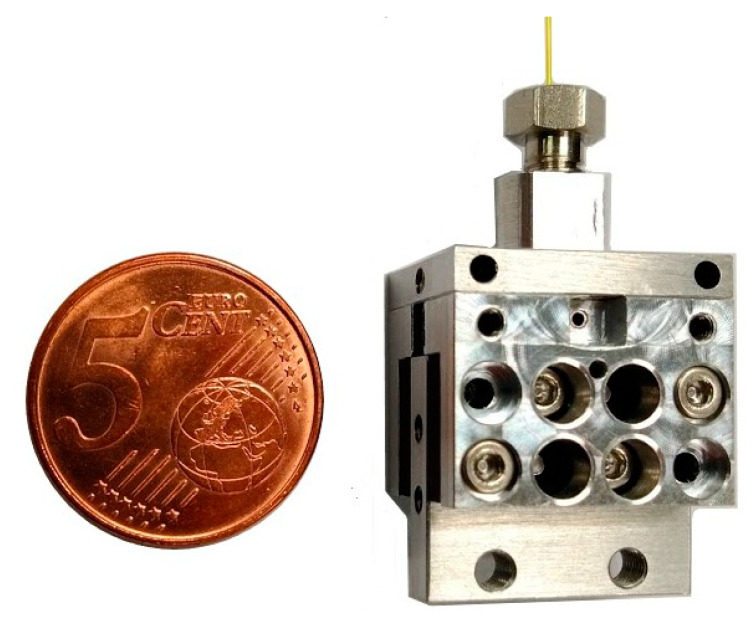
Miniaturized QEPAS cell.

**Figure 5 sensors-23-00270-f005:**
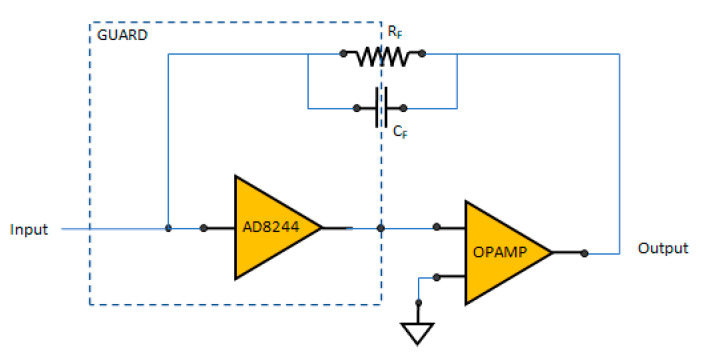
Composite amplifier scheme.

**Figure 6 sensors-23-00270-f006:**
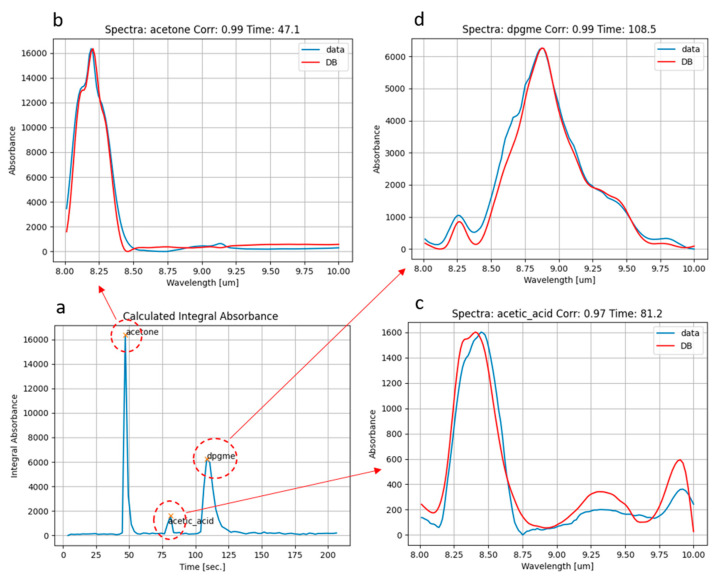
Total absorption chromatogram (**a**) obtained for a sampled mix of acetone and DPGME; spectra measured (blue line) and corresponding best fit from database (red line) of acetone (**b**) and DPGME (**d**) at the corresponding peaks of integral absorbance. A third intermediate peak was observed, whose corresponding measured and best-fit database spectra (**c**) correspond to acetic acid.

**Figure 7 sensors-23-00270-f007:**
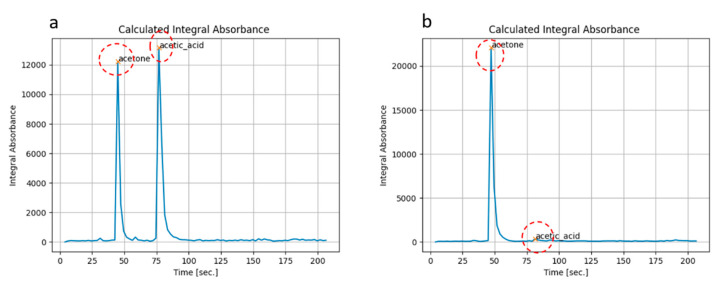
Total absorption chromatograms obtained by desorbing in air at 280 °C (**a**) or at 180 °C (**b**) vapors of acetone by means of the first-stage pre-concentrator, showing how high temperature could promote partial conversion of acetone into acetic acid.

**Figure 8 sensors-23-00270-f008:**
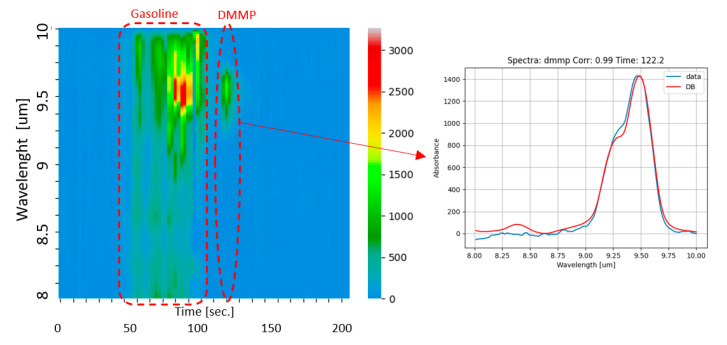
Color map (on the left) obtained by analyzing of a mix of saturated vapors of gasoline plus ~4 ppm of DMMP. The elution time is on the horizontal axis, the wavelength of acquired photoacoustic spectra is on the vertical axis, and the color bar is related to the intensity of the measured spectra. The spectrum acquired at 120 s, allowing one to identify the DMMP by comparison with the database, is plotted on the right.

**Figure 9 sensors-23-00270-f009:**
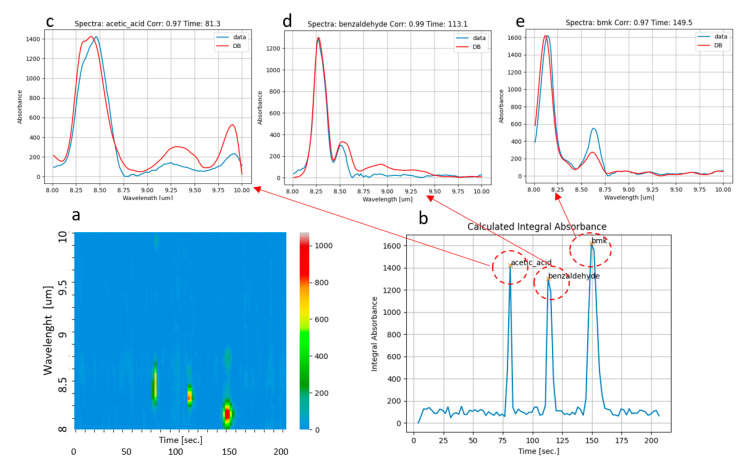
Analysis of BMK. (**a**): color map obtained as described in Figure 8; (**b**): the corresponding total absorption chromatogram; (**c**–**e**): spectra acquired at ca. 80 s, 115 s, and 150 s, allowing one to identify acetic acid, benzaldehyde, and BMK, respectively.

**Figure 10 sensors-23-00270-f010:**
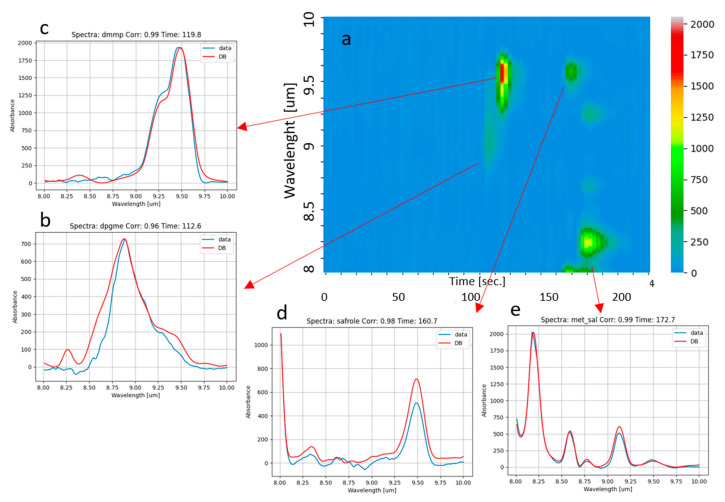
Analysis of a mix of DMMP, DPGME, methyl salicylate, and safrole. (**a**): color map obtained as described in Figure 8; (**b**–**e**): spectra acquired at ca. 110 s, 120 s, 160 s, and 170 s, allowing one to identify DMMP, DPGME, safrole, and methyl salicylate, respectively.

**Table 1 sensors-23-00270-t001:** Working parameters of the 1st stage pre-concentrator.

Sorbent materials	Carbograph 2TD and Carbograph 5TD
Sampling time	10–60 s
Sampling rate	800–1000 mL/min
Max desorption temperature	Up to 300 °C
Heating rate for desorption	5 °C/s

**Table 2 sensors-23-00270-t002:** Working parameters of the Compact-GC module.

MEMS Pre-Concentrator	
Sorbent Materials	Carbograph 2TD and 5TD
Sampling Time	10–60 s
Sampling Rate	200–400 mL/min
Max Temperature	Up to 300 °C
Heating Rate	20 °C/s
**MEMS Injector**	
Temperature	120–140 °C
Injection time	20–60 s
**MEMS Column**	
Stationary phase packing	Carbograph 1, coated
Start Temperature	60–80 °C
Hold Time at start T	5–30 s
Heating Rate	120–140 °C/min
Max Temperature	240–270 °C
Hold time at max T	1–2 min

**Table 3 sensors-23-00270-t003:** Working parameters of the QEPAS analyzer.

ADM	
QTF	Commercial OEM
Micro-resonator length	4.4 mm
Micro-resonator I.D.	0.9 mm
Temperature	80–120 °C
Internal Volume	~5 µL
**EC-QCL**	
Wavelength Range	8–10 µm
Amplitude Modulation	32,760−32,768 Hz
Pulse Duration	200 ns
Wavelength scan speed	0.13 cm^−1^/ms

## Data Availability

The data presented in this study are available on request from the corresponding author. The data are not publicly available due to the fact that RISEN program is still ongoing.

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
