# Peer review of "Compact GC-QEPAS for On-Site Analysis of Chemical Threats"

_sensors, 2022, doi:10.3390/s23010270_

Round 1

Reviewer 1 Report

The paper reports a compact, portable and selective chemical sensor for detecting hazardous vapors. The authors have proposed main upgrades of the sensor components and functional modules in detail along with test results.

The authors are requested to respond to the below comments point by point.

1)         The introduction should be enriched with the most recent research papers (last three to four years) on the proposed work.

2)         On page no:4, lines 102 and 103, The material used to construct MEMS Injector and MEMS GC separation column?

3)         The approximate power consumption (in watts) of the compact GC prototype specified in Figure 3 and also the carrier gas used in the GC column.

4)         Please specify the structure of the QTF and microresonators specified in Table 3.

5)         What is the full form of DPGME? The Y-axis label for Figure 6C is missing.

6)         Do the authors fabricate the QTF used in the QEPAS detector?

7)         What is the pressure range of the Absolute Integrated Pressure Sensor (model MPXH6400A)?

8)         Please explain in detail the working of QEPAS in detecting the volatile and semi-volatile compounds.

9)         The authors should compare their work with a similar kind of work in the literature which can be included in the revised version.

Reviewer 2 Report

Dear authors,

congratulations to an interesting approach and great results. I have read your paper and I have attached a review PDF including my remarks/questions. Although it looks like many questions, most of them can be answered simply and the information I would like you to add to the publication is manageable.

Besides, I have attached a second PDF with your publication. I have highlighted sentences or passages where I suggest a revision of the language or where I found typos. If you cannot see my remarks in the PDF - I use "PDF-XChange", which is freeware.

I am looking forward to your answers.

Reviewer 3 Report

Comments:

1. Why particularly a graphitized carbon sorbent mesh for capturing substances is used in the sampling unit? How is it more efficient than the other

2. There are a few grammatical errors in line no. 132, 139, 140 (para-2.3.1). In the same para, line no. 137, the spelling of “tacking” should be corrected to “taking”.

3. In line no. 170 (para 2.4) “lockin” should be written as “lock-in”.

4. Why does GC need two separate Nitrogen pressure inlets? Please give a short explanation for this.

5. In line no. 257 (para 3.2) “sampled” should be replaced with “sample”.
